# impHFrEF trial: study protocol for an open-label, multicentre study of improvement the outcome of patients with heart failure in China using a mobile hEalth-supported platForm

Fuzhong Chen [1], Guangjuan Li,[1,2] Xinxin Zhang,[3] Qin Shen,[1] Fangfang Wang,[4] Xiaoyu Dong,[1] Yu Zou,[1] Wensen Chen,[5,6] Bing Xu,[3] Junhong Wang[1,7]

FC, GL and XZ contributed equally.

For numbered affiliations see end of article.

**Correspondence to**
Professor Junhong Wang;
wangjunhong@jsph.org.cn, Dr Bing Xu;
nyxb1980@163.com and Dr Wensen Chen;
wensenchen@njmu.edu.cn

## ABSTRACT

**Background** Patients with chronic heart failure (CHF) often have a long duration of illness, difficulty in attending follow-up visits, and poor adherence to treatment. As a result, they frequently cannot receive guideline-directed medical therapy (GDMT) at the desired or maximum tolerable drug dosage. This leads to high hospitalisation and mortality rates for HF patients. Therefore, effective management and monitoring of patients with HF to ensure they receive GDMT is crucial for improving the prognosis.

**Design and methods** This is a multicentre, open-label, randomised, parallel-group study involving patients with CHF across five centres. The study aims to assess the impact of an optimised GDMT model for HF patients, established on a mobile health (mHealth) platform, compared with a control group. Patients must have a left ventricular ejection fraction of less than 50% and be receiving medication titration therapy that has not yet reached the target dose, with a modest increase in N-terminal pro-B-type natriuretic peptide level. The primary composite outcome is worsening HF events (hospitalisation or emergency treatment with intravenous fluids) or cardiovascular death.

**Ethics and dissemination** On 22 December 2021, this study received ethical approval from the Ethics Review Board of the First Affiliated Hospital of Nanjing Medical University, with the ethics number 2021-SR-530. All study participants will be informed of the research purpose and their participation will be voluntary. Informed consent will be obtained by providing and signing an informed consent form. We will ensure compliance with relevant laws and regulations regarding privacy and data protection. The results of this study will be published in a peer-reviewed academic journal. We will ensure that the dissemination of study results is accurate, clear and timely.

**Trial registration number** ChiCTR2200056527.

## STRENGTHS AND LIMITATIONS OF THIS STUDY

⇒ The study followed ethical principles and legal regulations, ensuring the safety and legitimacy of the research.
⇒ As a randomised controlled prospective clinical trial with a large sample size, the study provided reliable results.
⇒ As a prospective clinical study, it is not possible to fully control all variables, which may have an impact on the accuracy of the research results.
⇒ The generalisability and applicability of the research results may be limited due to variations in medical standards across countries and regions.

factors, leading to impaired ventricular systolic and/or diastolic function.[1 2] The main manifestations are dyspnoea, fatigue and fluid retention (pulmonary circulation congestion, systemic circulation congestion and peripheral oedema).[1 3] Most patients with acute HF have partial remission of symptoms after hospitalisation and are referred to CHF; patients with CHF often require hospitalisation due to acute exacerbation of various triggers. HF has become an important public health issue worldwide due to its high incidence, rehospitalisation rate and death rate.[4 5] China-HF study shows a 4.1% mortality rate for hospitalised patients with HF in China.[5] The 2012–2015 Chinese Epidemiological Survey showed that the prevalence of HF among adults over 35 years old in China was 1.3% (estimated 13.7 million).[6] As China's population ages and the incidence of chronic diseases such as coronary heart disease (CHD), hypertension and diabetes increases, improvements in medical care have led to longer survival periods for patients with heart disease, resulting in a continuing trend of increased HF prevalence in China.

## INTRODUCTION

The term chronic heart failure (CHF) refers to a complex clinical syndrome resulting from abnormal changes in the structure and/or function of the heart owning to numerous

BMJ

Guideline-directed medical therapy (GDMT) is a cornerstone of care for patients with CHF. In terms of drug therapy, ACE inhibitors (ACEIs) or angiotensin II receptor antagonists (ARBs), beta-blockers (BBs) and mineralocorticoid receptor antagonists (MRAs) constitute the standard therapy for the treatment of CHF. This can significantly improve the clinical symptoms of patients with CHF and reduce the rate of rehospitalisation and mortality. Over the past few years, a variety of new HF therapeutic agents have emerged that can improve prognosis. Among them, extensive evidence of the clinical benefit of angiotensin receptor-neprilysin inhibitors (ARNIs) and sodium-glucose co-transporter-2 inhibitors (SGLT2is) has led to a shift from the 'ACEI/ARB+BB+MRA' regimen to the 'ARNI/ACEI/ARB+SGLT2i+BB+ MRA' regimen for improving the prognosis of patients with HF with reduced ejection fraction.[3 7–9]

Clinical guidelines emphasise the importance of achieving the target dose of GDMT as much as possible.[3 7] However, due to the long duration of HF, it is difficult for patients to be rechecked and examined after discharge due to medical and economic constraints. In addition, severe side effects such as sinus bradycardia, atrioventricular block and blood pressure reduction can occur during β-blockers, ACEIs/ARBs and ARNIs therapy, leading to exacerbation and recurrence of HF. As a result, patients' adherence to treatment is poor after discharge from the hospital, making it impossible for patients to receive the target dose of medication. This ultimately leads to high rates of rehospitalisation and death from HF. Recent studies have shown that even in the medically advanced USA, the vast majority of patients with HF are not titrated to the target dose or the maximum tolerated dose of BBs and ACEIs/ARBs.[10–12] Therefore, effective follow-up and management of patients are key to improving the prognosis of patients with HF and saving healthcare resources.

Recent studies in China and abroad have attempted to use existing social media tools such as WeChat, Facebook and Twitter to manage patients with CHD and other cardiovascular diseases to effectively reduce the incidence of cardiovascular events in patients with CHD. Dorje T *et al* reported in Lancet on the successful use of a WeChat social platform to guide the treatment of patients with CHD through refined management for rehabilitation and secondary prevention.[13] As a result, an APP has been developed for patients with CHF based on the mobile health (mHealth) platform. The APP can be used to monitor the daily condition of the patient, allowing the physician and patient to communicate in real-time through the software. This establishes a remote monitoring platform with the participation of doctors and patients. By monitoring the arrhythmias of patients with HF and their responsiveness to medication, patients can be effectively monitored remotely and guided in the use of medication. Ultimately, patients with HF will be effectively managed in a timely and scientific manner, reducing the number of repeated hospitalisations, lowering the death rate of HF patients, and reducing the medical burden on society.

Consequently, the implementation of this study can establish an optimal GDMT management model for patients with HF based on the mHealth platform. This model can be effectively promoted and popularised, ultimately improving the prognosis of patients with HF and saving social medical resources.

## METHODS
The study is a multicentre, parallel-group, randomised, open-label, prospective trial in patients with CHF. The study is scheduled to commence in June 2022 and is expected to conclude in December 2024. The registration number of the study is ChiCTR2200056527.

### Study design and conduct
#### Trial design overview
The study is a multicentre, prospective, open-label study. A total of five centres participated in this study: Jiangsu Provincial People's Hospital, Northern Jiangsu People's Hospital, the Second Affiliated Hospital of Nanjing Medical University, Changzhou No.2 People's Hospital and the Friendship Hospital of Ili Kazak Autonomous Prefecture. In each centre, a team of approximately 2–3 clinical physicians is engaged, with the same doctors overseeing both trial and non-trial patients. In addition, a seasoned cardiologist conducts monthly quality control assessments at each centre to ensure the rationality of drug treatment adjustments and the integrity of follow-up visits, thus ensuring the homogeneity of the study. A total of 680 patients with CHF are planned to be randomly enrolled and assigned 1:1 to the mhealth optimised GDMT group (mGDMT group) and the control group, with 340 patients in each group. The population included is HF medication not titrated to the target doses or maximum tolerated doses. Patients in the mGDMT group are managed individually and precisely through a mHealth platform application (app), and the dosage of BBs and other HF treatment medications are titrated and adjusted to the patient's tolerated target dose according to the results of the remote mHealth platform. The control group receives the current clinical routine follow-up and treatment. In the control group, patients will be treated by experienced cardiovascular internists at the hospital, who will adjust their treatment plans according to the Chinese guidelines for HF. The frequency of clinic visits for these patients will initially be set at 2–4 weeks, but could be extended to once every 1–3 months for those who have reached the target medication dose and have stabilised their condition. Follow-up will be performed at the intervals of 1 month, 3 months, 6 months and 12 months

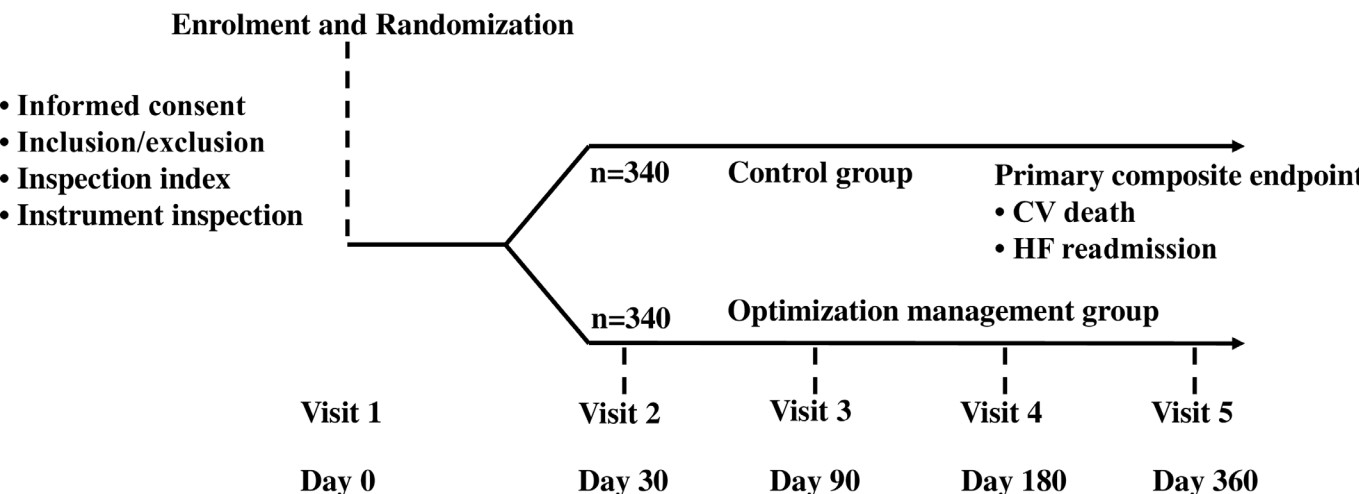

**Figure 1** Trial design. CV, cardiovascular; HF, heart failure.

after enrolment for a total of 12 months to compare clinical symptoms, cardiac function classification, incidence of adverse events, hospitalisation rate, mean number of days in hospital and mortality between the two groups. Figure 1 illustrates the trial procedures of the study.

### The mHealth platform app construction

The mHealth platform App builds an interactive platform for doctors and patients to participate together. The App includes a doctor-side and a patient-side portal. The patient side of the software requires the patient to bind to the appropriate doctor and clock in daily on their weight, urine output, blood pressure, heart rates, HF symptoms and other indicators (laboratory results or examination for local review). The patient-side data are uploaded and presented in a visualised way on the doctor side. The physician understands the patients' medication reaction based on the data uploaded daily by the patient and communicates effectively with the patient through the App platform dialogue tool. The corresponding physician makes precise and standardised adjustments to the HF medication and treatment according to the patients' condition (by using electronic prescriptions). At the same time, the platform sets up an alarm function for extreme values to alert physicians when abnormal values of key indicators appear in patients, facilitating timely adjustment of patients' medication regimens. Throughout the visit, the importance of patient adherence with programme requirements regarding daily measurements and symptom reporting is emphasised to ensure timely identification of potential changes in their condition. Figure 2 demonstrates the model for optimal management of patients with HF based on a mHealth App.

### Patients

The population included in this study is patients with HF who have new or ongoing HF medication that has not yet reached target doses and requires adjustment of medication to target or goal measures, or patients with HF who have been treated to stop the intravenous vasoactive medication and then switched to oral medication after stabilisation. The patient's left ventricular ejection fraction as measured by the Simpson method needs to be <50% and the New York Heart Association (NYHA) classes II–IV (NYHA classes IV in stable phase). The patient's brain natriuretic peptide (BNP) concentration should be ≥150 pg/mL or N-terminal pro-B-type natriuretic peptide (NT-proBNP)≥600 pg/mL, or BNP≥100 pg/mL if the patient has been hospitalised for HF within 12 months (NT-proBNP≥400 pg/mL). Subjects must meet the age limit of 18 years, be proficient in the use of electronic devices (proficiency in using WeChat app). Figure 3 displays the inclusion criteria for the trial.

The key exclusion criteria for the impHFrEF trial include somatic activity disorders and psychiatric disorders, chronic obstructive pulmonary disease and asthma, malignant tumour patients with a life expectancy of less than 1 year, plans to receive device or surgical treatment such as a heart transplant, left ventricular assist device, CRTD (Cardiac resynchronization therapy defibrillator) implantation or valve replacement surgery during the follow-up period, recurrent episodes of intractable HF, various correctable secondary causes of HF including hyperthyroidism, anaemic heart disease, congenital heart disease, organic valvular heart disease. Patients with renal failure (creatinine clearance <30 mL/min) or on dialysis treatment, hyperkalemia (blood potassium >5.5 mmol/L) at the time of enrolment and those who are participating in other clinical trials are also excluded. Fgure 4 presents the exclusion criteria for the trial.

On providing their informed consent, eligible patients who consent to participate in this clinical trial will be randomly assigned to their respective groups.

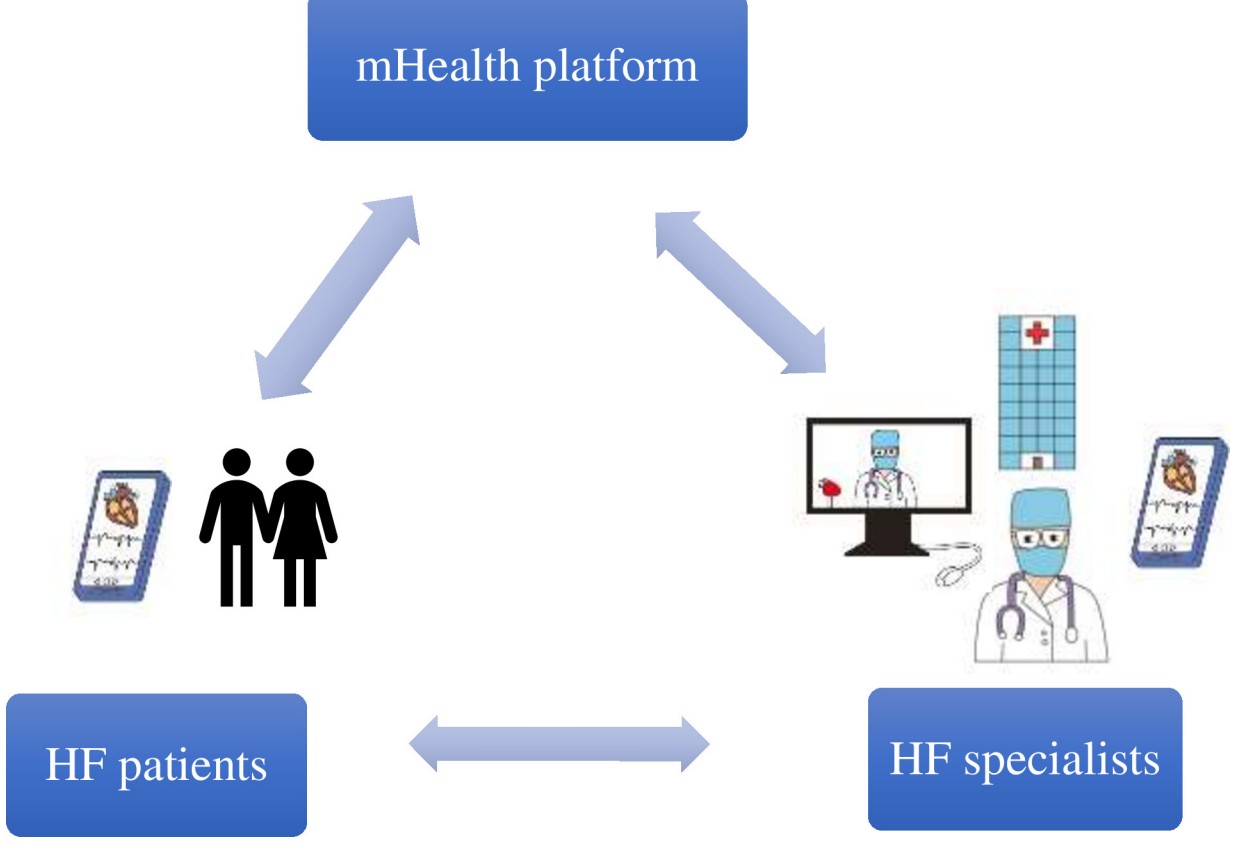

**Figure 2** Optimised GDMT model for managing HF patients based on the mHealth platform. GDMT, guideline directed medical therapy; HF, heart failure; mHealth, mobile health.

All patient data shall be held in strict confidence and accessible exclusively by authorised personnel. The usage of patients' personal information will be strictly limited to research purposes only. Under no circumstances shall patient information be used for any other purposes without explicit written consent from the individuals involved. At the conclusion of the investigation, all patient data will undergo meticulous anonymisation procedures, guaranteeing the utmost protection of patient confidentiality.

---

- Willing to provide written informed consent

- 18 years or older

- Left ventricular ejection fraction (LVEF) as measured by the Simpson method needs to be <50% and the New York Heart Association (NYHA) Classes II-IV (NYHA IV in stable phase)

- The patient's brain natriuretic peptide (BNP) concentration should be ≥150 pg/ml or N-terminal pro-B-type natriuretic peptide (NT-proBNP) ≥600 pg/ml, or BNP ≥100 pg/ml (NT-proBNP ≥400 pg/ml) if the patient has been hospitalized for heart failure within 12 months

- Heart failure patients with new or ongoing anti-heart failure medication that has not yet reached target doses and requires adjustment of medication to target or goal measures, or heart failure patients who have been treated to stop the intravenous vasoactive medication and then switched to oral medication after stabilization

- Patients should be proficient in the use of electronic devices

---

**Figure 3** Patient inclusion criteria.

- Somatic Activity Disorders and Psychiatric Disorders

- Chronic obstructive pulmonary disease (COPD) and Asthma

- Malignant tumor patients (life expectancy < 1 year)

- Plans to receive device or surgical treatment (such as a heart transplant, left ventricular assist device or CRTD implantation and valve replacement surgery during the follow-up period)

- Recurrent episodes of intractable heart failure, Various correctable secondary causes of heart failure (including hyperthyroidism, anemic heart disease, congenital heart disease, organic valvular heart disease)

- Patients with renal failure (creatinine clearance <30 ml/min) or on dialysis treatment

- Hyperkalemia (blood potassium >5.5 mmol/l) at randomization

- Those who are participating in other clinical trials

**Figure 4** Patient exclusion criteria. CRTD, cardiac resynchronization therapy defibrillator.

## Randomisation

Once patients meet the criteria for randomisation, they are assigned by investigators through a central randomisation system. Patients in the study are centrally randomised and competitively enrolled in either the mGDMT group or the normal control groups.

## Study visits and follow-up

Patients who meet the inclusion criteria will be evaluated on admission and randomly assigned to either the mGDMT group or the control group after signing an informed consent form. Baseline information, including clinical examination and laboratory measurements, will be collected. Follow-up visits will be conducted on days 30±3, 90±3, 180±3 and 360±3 after randomisation. These visits will focus on HF symptoms, signs, volume status and adverse events. ECG, echocardiography and laboratory results such as troponin, pro-BNP, liver and kidney function will also be reviewed. The schedule for data acquisition is shown in table 1.

## Outcomes

The primary endpoint is a composite endpoint consisting of cardiac death (deaths due to acute myocardial infarction, HF, systemic embolism and severe arrhythmic events) and hospitalisation due to worsening HF symptoms. Worsening HF symptoms are defined as the presence of HF that cannot be controlled by oral medications and requires hospitalisation or emergency room treatment with intravenous medications.

The secondary endpoints primarily include (1) all-cause death; (2) cumulative HF admissions during the trial; (3) changes in the 6 min walk test and HF biomarkers and (4) changes from baseline to 12 months in the total symptom score using the Kansas City Cardiomyopathy Questionnaire.[14]

The study is expected to collect as much clinical information as possible from patients during each visit. This information will be used to compare clinical symptoms, cardiac function class, incidence of adverse events, hospitalisation rate, mean days in the hospital and mortality between the two groups of patients. The efficacy of HF medication and adverse events will be monitored and recorded.

## Statistical considerations
### Sample size assumptions

The incidence rate of endpoint events in the control group was determined based on previous research findings. The annual event rate of HF recurrence, cardiogenic and all-cause death in the control group is 15%.[15] The mGDMT group expected an annual event rate of 8%.[16] With a test efficacy of 0.8 and a bilateral alpha of 0.05, a sample size ratio of 1:1 for inclusion of the control and mGDMT groups, and a lost-to-review rate of 5%, a minimum sample size of 340 cases is required for each of the two groups. Therefore, the study expects to include a total of 680 patients.

### Methods for statistical analyses

In our experiment, the statistical methods have been meticulously preplanned. All analyses will be conducted using R (V.4.2.2) and SPSS (V.27), both of which are robust and widely accepted statistical software packages. The primary and secondary efficacy analyses will be performed on the full analysis set, adhering to the principle of intention to treat. This principle ensures that the results are not biased due to postrandomisation exclusions. The full-analysis set will include all randomised patients who have provided informed consent and have commenced the indicated treatment.

**Table 1** Schedule for data acquisition indicated by checkmarks at the specified time point

| Data collected | Baseline | Day 30 | Day 90 | Day 180 | Day 360 |
|---|---|---|---|---|---|
| Demographics | ✓ | N/A | N/A | N/A | N/A |
| History | ✓ | N/A | N/A | N/A | N/A |
| Vital signs | ✓ | ✓ | ✓ | ✓ | ✓ |
| Signs and symptoms | ✓ | ✓ | ✓ | ✓ | ✓ |
| NYHA class | ✓ | ✓ | ✓ | ✓ | ✓ |
| 6 min walk test | ✓ | ✓ | ✓ | ✓ | ✓ |
| KCCQ scores | ✓ | ✓ | ✓ | ✓ | ✓ |
| Laboratory results | | | | | |
| BNP levels | ✓ | N/A | N/A | ✓ | ✓ |
| Troponin | ✓ | N/A | N/A | ✓ | ✓ |
| Renal function | ✓ | N/A | N/A | ✓ | ✓ |
| Liver function | ✓ | N/A | N/A | ✓ | ✓ |
| Examination results | | | | | |
| ECG | ✓ | N/A | N/A | ✓ | ✓ |
| Echocardiography | ✓ | N/A | N/A | ✓ | ✓ |
| Endpoints | | | | | |
| Cardiac death | N/A | ✓ | ✓ | ✓ | ✓ |
| Number of HF-related hospitalisations since enrollment | N/A | ✓ | ✓ | ✓ | ✓ |
| Number of days in the hospital since enrollment | N/A | ✓ | ✓ | ✓ | ✓ |
| All-cause death | N/A | ✓ | ✓ | ✓ | ✓ |

BNP, brain natriuretic peptide; HF, heart failure; KCCQ, Kansas City Cardiomyopathy Questionnaire; N/A, not applicable; NYHA, New York Heart Association.

For data conforming to a normal distribution, it will be expressed as mean±SD. An independent sample t-test, a parametric test, will be employed for comparison. For variables with a skewed distribution, they will be expressed as the median and IQR, and a nonparametric test will be used for comparison. This approach allows us to handle data that do not meet the assumptions of parametric tests. Categorical data will be presented as counts and percentages. $\chi^2$ or Fisher's exact tests will be used for evaluation, depending on the distribution and characteristics of the data. The Kaplan-Meier method, a non-parametric statistic used to estimate the survival function from lifetime data, will be used to assess the difference in the incidence of endpoint events between the two groups of patients. In addition, we will conduct a sensitivity analysis to assess the robustness of our findings. A two-sided $p<0.05$ will be considered statistically significant. All statistical tests will be two sided, reflecting our non-directional hypotheses.

Our statistical team, with their extensive experience and expertise, will ensure the accuracy and reliability of our analyses. By adhering to these rigorous statistical methods, we aim to produce precise and trustworthy experimental results.

### Ancillary studies
After obtaining informed consent from patients, blood samples will be collected at baseline and during follow-up for the measurement of biomarkers of interest.

### Trial design and governance
The study was designed by the Cardiovascular Medicine Department of the First Affiliated Hospital of Nanjing Medical University and reviewed by its Ethics Committee. The study has completed clinical trial registration and obtained a registration number. There are five study centres, and the study will be supervised by the ethics committees of their respective centres during the study implementation.

### Patient and public involvement
Patients or the public were not involved in the design, or conduct, or reporting, or dissemination plans of our research.

### Safety considerations
Patients with HF often have a changing condition, with mortality and rehospitalisation rates as high as 15% and 30% within 2–3 months of their discharge from the hospital.[17] Therefore, investigators need to evaluate patients' uploaded heart rates, blood pressure, HF symptoms and test results to adjust therapeutic drug doses according to their haemodynamic status and fluid retention. If patients develop renal insufficiency or hyperkalaemia during treatment, the medication should also be adjusted immediately according to the guidelines.[3] The trial should be terminated if there is a rapid deterioration in renal function (50% increase in creatinine level or 0.3 mg/dL increase in absolute creatinine level compared with 48 hours before) or severe sinus bradycardia requiring pacing therapy.

### Current status
The First Affiliated Hospital of Nanjing Medical University ethically cleared the study on 22 December 2021, and clinical trial registration was completed on 7 February 2022. Ethics number is 2021-SR-530. The mHealth platform APP was constructed on 1 June 2022. The first subject was recruited and randomised to groups on 29 June 2022. A later phase will rapidly recruit subjects and complete randomisation.

## SUPPLEMENTARY INFORMATION
### Ethics and dissemination

On 22 December 2021, this study received ethical approval from the Ethics Review Board of the First Affiliated Hospital of Nanjing Medical University, with the ethics number 2021-SR-530. All study participants will be informed of the research purpose and their participation will be voluntary. Informed consent will be obtained by providing and signing an informed consent form. We will ensure compliance with relevant laws and regulations regarding privacy and data protection. The results of this study will be published in a peer-reviewed academic journal. We will ensure that the dissemination of study results is accurate, clear and timely.

**Author affiliations**
¹Department of Cardiology, The First Affiliated Hospital with Nanjing Medical University, Nanjing, Jiangsu, China
²Department of Cardiology, The Friendship Hospital of Ili Kazak Autonomous Prefecture, Yining, China
³Department of Cardiology, Northern Jiangsu People's Hospital, Yangzhou, Jiangsu, China
⁴Department of Cardiology, Changzhou No.2 People's Hospital, Changzhou, Jiangsu, China
⁵Office of Infection Management, The First Affiliated Hospital with Nanjing Medical University, Nanjing, Jiangsu, China
⁶Department of Epidemiology and Biostatistics, School of Public Health, Xi'an Jiaotong University Health Science Center, Xi'an, Shanxi, China
⁷Jiangsu Health Administration and Development Research Center, Nanjing, China

**Contributors** FC contributed to the randomisation and grouping of patients, as well as the writing and revision of the paper. GL was responsible for patient recruitment and contributed to the paper revision. XZ, QS and FW were involved in patient recruitment, data collection and collation. XD assisted in patient randomisation, grouping and data collection. YZ supported data collection and collation. WC provided statistical expertise and guidance on trial design. BX offered guidance on trial design and supervised its implementation. JW provided comprehensive guidance on trial design and implementation, oversaw and coordinated the entire research process, and funded the study. Each author played a critical role in ensuring the accuracy and completeness of the research findings and is accountable for the final content of the paper.

**Funding** This work was supported by the National Natural Science Foundation of China (NSFC 82170269), Jiangsu Province Hospital (the first affiliated hospital with Nanjing Medical University) clinical capacity enhancement project (JSPH-MB-2021-15), the Foundation of Clinical Medical Research Center of Yili Autonomous Prefecture (YL2020ms09) and the open project of Jiangsu health administration and development research center (JSHD2022009).

**Competing interests** None declared.

**Patient and public involvement** Patients and/or the public were not involved in the design, or conduct, or reporting, or dissemination plans of this research.

**Patient consent for publication** Not applicable.

**Provenance and peer review** Not commissioned; externally peer reviewed.

**ORCID iD**
Fuzhong Chen http://orcid.org/0009-0009-5054-0439

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
