## [Reviewer comments · BMJ Open]

ARTICLE DETAILS

TITLE (PROVISIONAL)	The impHFrEF trial: study protocol for an open-label, multi-center study of improvement the outcome of patients with Heart Failure in China using a mobile hEalth-supported platForm
AUTHORS	Chen, Fuzhong; Li, Guangjuan; Zhang, Xinxin; Shen, Qin; Wang, Fangfang; Dong, Xiaoyu; Zou, Yu; Chen, Wensen; Xu, Bing; Wang, Junhong

VERSION 1 – REVIEW

REVIEWER	Pallath , Vinod Universiti Malaya
REVIEW RETURNED	10-Nov-2023

GENERAL COMMENTS	1. No ethics approval details are given in the protocol2. The protocol does not contain procedures for informed consent and patient confidentiality details3. The description of sample size calculation is inadequate4. The description of statistical analysis is not given5. The outcomes stated are not explicit, can be described better for reproducibility6. Strengths and limitations of the study protocol not provided.
--

REVIEWER	Shah, Anoop University College London, Farr Institute of Health Informatics Research
REVIEW RETURNED	12-Dec-2023

GENERAL COMMENTS	This is a study protocol for a trial of a mobile health platform to assist in guideline directed medical therapy in chronic heart failure. This is a multicentre, parallel-group, randomized, open-label, prospective clinical trial. The primary endpoint is a composite endpoint consisting of cardiac death and hospitalization due to worsening heart failure symptoms. General comments: The trial is comparing a care model involving a mobile health platform with routine care. The trial aims, inclusion and exclusion criteria and outcomes are described clearly. However, the intervention itself (the clinical care service involving the mobile app and routine care) are not described in sufficient detail for a reader to understand what is actually being compared, and whether this intervention will work in other countries or settings.
---

	The protocol needs to include information about how frequently clinicians will check the data entered by each patient and how they will alter the medication (e.g. do they issue prescriptions or advise the patient doses of drugs they already have). How many clinicians will be involved? Will the same clinicians be looking after some trial patients and some non-trial patients? The type of care received in the control arm also needs to be specified - e.g. frequency of visits or phone calls, whether in a hospital or clinic or general practice, and type of clinician (e.g. doctor or specialist nurse). How are medication changes usually carried out, and what guidelines are usually followed? Do patients need to pay for their medications in each arm of the trial, and does cost influence what medications a patient chooses to take? It would be important to collect and compare data on the amount of clinician time involved in managing patients in each arm of the trial, as this might be a possible explanation for any differences in outcomes. Minor comments: Page 9 9line 23-25 - what is 'daily punch card data'? Page 6 line 56 - why is APP capitalised? Page 10 - the exclusion criteria do not need to start with capital letters
--	--

VERSION 1 – AUTHOR RESPONSE

Reviewer: 1

Comments to the Author:

1. No ethics approval details are given in the protocol

Responses: Thank you for the reminder. We have now included details of the ethical approval in both the abstract and the main body of the manuscript.

2. The protocol does not contain procedures for informed consent and patient confidentiality details

Responses: We feel sorry that we did not clarify this issue in the present manuscript. We have now included information regarding procedures for informed consent and patient confidentiality in the protocol. Thank you for bringing this to our attention.

3. The description of sample size calculation is inadequate

Responses: We appreciate the suggestion for providing more details on the sample size calculation. The incidence rate of endpoint events in the control group was determined based on previous research findings. The annual event rate of heart failure recurrence, cardiogenic and all-cause death in the control group is 15%. The mGDMT group expected an annual event rate of 8%. With a test efficacy of 0.8 and a bilateral alpha of 0.05, a sample size ratio of 1:1 for inclusion of the control and mGDMT groups, and a lost-to-review rate of 5%, a minimum sample size of 340 cases is required for each of the two groups. Therefore, the study expects to include a total of 680 patients.

$$650 * (1 + 0.05) \approx 680$$

4. The description of statistical analysis is not given

Responses: We apologize for the lack of clarity in our statistical description.

In our experiment, the statistical methods have been meticulously preplanned. All analyses will be conducted using R (version 4.2.2) and SPSS (version 27), both of which are robust and widely accepted statistical software packages. The primary and secondary efficacy analyses will be

performed on the full analysis set, adhering to the principle of intention-to-treat. This principle ensures that the results are not biased due to post-randomization exclusions. The full analysis set will include all randomized patients who have provided informed consent and have commenced the indicated treatment.

For data conforming to a normal distribution, it will be expressed as mean \pm standard deviation. An independent samples t-test, a parametric test, will be employed for comparison. For variables with a skewed distribution, they will be expressed as the median and interquartile range (IQR), and a nonparametric test will be used for comparison. This approach allows us to handle data that do not meet the assumptions of parametric tests. Categorical data will be presented as counts and percentages. Chi-square or Fisher exact tests will be utilized for evaluation, depending on the distribution and characteristics of the data. The Kaplan-Meier (K-M) method, a non-parametric statistic used to estimate the survival function from lifetime data, will be used to assess the difference in the incidence of endpoint events between the two groups of patients. In addition, we will conduct a sensitivity analysis to assess the robustness of our findings. A two-sided p-value less than 0.05 will be considered statistically significant. All statistical tests will be two-sided, reflecting our non-directional hypotheses.

Our statistical team, with their extensive experience and expertise, will ensure the accuracy and reliability of our analyses. By adhering to these rigorous statistical methods, we aim to produce precise and trustworthy experimental results.

5. The outcomes stated are not explicit, can be described better for reproducibility

Responses: We are very sorry that we have not made it clear in the manuscript. We have now provided a revised explanation of the primary and secondary outcomes. The primary endpoint is a composite endpoint consisting of cardiac death (deaths due to acute myocardial infarction (AMI), HF, systemic embolism, and severe arrhythmic events) and hospitalization due to worsening heart failure symptoms. Worsening heart failure symptoms are defined as the presence of heart failure that cannot be controlled by oral medications and requires hospitalization or emergency room treatment with intravenous medications. The secondary endpoints primarily include : (i) all-cause death; (ii) cumulative heart failure admissions during the trial; (iii) changes in the 6-minute walk test and heart failure biomarkers; (iv) changes from baseline to 12 months in the total symptom score using the Kansas City Cardiomyopathy Questionnaire (KCCQ).

6. Strengths and limitations of the study protocol not provided.

Responses: We feel sorry that we did not clarify this point in the manuscript. We have now added a section on the strengths and limitations of the study protocol after the abstract.

Reviewer: 2

However, the intervention itself (the clinical care service involving the mobile app and routine care) are not described in sufficient detail for a reader to understand what is actually being compared, and whether this intervention will work in other countries or settings.

The protocol needs to include information about how frequently clinicians will check the data entered by each patient and how they will alter the medication (e.g. do they issue prescriptions or advise the patient doses of drugs they already have). How many clinicians will be involved? Will the same clinicians be looking after some trial patients and some non-trial patients?

Responses: We are grateful for the thoughtful suggestions provided by the reviewers, and have made the necessary modifications to the article. The physician understands the patients' medication reaction based on the data uploaded daily by the patient and communicates effectively with the patient through the App platform dialogue tool. The corresponding physician makes precise and

standardized adjustments to the heart failure medication and treatment according to the patients' condition (by utilizing electronic prescriptions). In each center, a team of approximately 2-3 clinical physicians is engaged, with the same doctors overseeing both trial and non-trial patients. In addition, a seasoned cardiologist conducts monthly quality control assessments at each center to ensure the rationality of drug treatment adjustments and the integrity of follow-up visits, thus ensuring the homogeneity of the study. However, due to the varying levels of medical care across different countries, we cannot guarantee this intervention is applicable in all countries or settings.

The type of care received in the control arm also needs to be specified - e.g. frequency of visits or phone calls, whether in a hospital or clinic or general practice, and type of clinician (e.g. doctor or specialist nurse). How are medication changes usually carried out, and what guidelines are usually followed?

Responses: Thank you very much for your reminder. We have added some details about the treatment of patients in the control group. In the control group, patients will be treated by experienced cardiovascular internists at the hospital, who will adjust their treatment plans according to the Chinese guidelines for heart failure. The frequency of clinic visits for these patients will be asked to set at every 2-4 weeks, but could be extended to once every 1-3 months for those who have reached the target medication dose and have stabilized their condition.

Do patients need to pay for their medications in each arm of the trial, and does cost influence what medications a patient chooses to take?

Responses: We feel sorry that we did not clarify this point in the manuscript. In both the intervention and control groups, patients are required to bear the cost of purchasing medications. However, in China, the treatment medications for heart failure are covered by medical insurance, so the cost is unlikely to influence patients' choice of which medication to take.

It would be important to collect and compare data on the amount of clinician time involved in managing patients in each arm of the trial, as this might be a possible explanation for any differences in outcomes.

Responses: We highly appreciate and completely agree with the reviewer's thoughtful comments. We will collect and compare the amount of time spent by clinicians in the intervention and control groups, and investigate whether this could contribute to any differences in outcomes.

Minor comments:

Page 9 line 23-25 - what is 'daily punch card data'?

Responses: We are sorry for this confusion. "Punch card data" refers to the daily data uploaded by patients, including symptoms and vital signs. To ensure clarity, we have revised the sentence to: "The physician get the patients' medication information based on the data uploaded daily by the patient and communicates effectively with the patient through the App platform dialogue tool. "

Page 6 line 56 - why is APP capitalised?

Responses: We agree with the reviewer's comment and have changed "APP" to lowercase "app".

Page 10 - the exclusion criteria do not need to start with capital letters

Responses: Thank you for your suggestion. We have made the change from "Exclusion criteria" to "exclusion criteria".

We sincerely appreciate the time and thoughtful comments and recommendations by the editor and reviewers. We believe we adequately addressed the questions to the best of our ability.

VERSION 2 – REVIEW

REVIEWER	Shah, Anoop University College London, Farr Institute of Health Informatics Research
REVIEW RETURNED	29-Jan-2024
GENERAL COMMENTS	I am satisfied that the authors have made the recommended changes to the manuscript.

VERSION 2 – AUTHOR RESPONSE